# New Insights in the Occurrence of Venous Thromboembolism in Critically Ill Patients with COVID-19—A Large Postmortem and Clinical Analysis

**DOI:** 10.3390/v14040811

**Published:** 2022-04-14

**Authors:** Fabian Heinrich, Kevin Roedl, Dominik Jarczak, Hanna-Lisa Goebels, Axel Heinemann, Ulrich Schäfer, Frank Ludwig, Martin Bachmann, Berthold Bein, Christian Friedrich Weber, Karsten Sydow, Marc Bota, Hans-Richard Paschen, Andreas de Weerth, Carsten Veit, Oliver Detsch, Philipp-Alexander Brand, Stefan Kluge, Benjamin Ondruschka, Dominic Wichmann

**Affiliations:** 1Institute of Legal Medicine, University Medical Center Hamburg-Eppendorf, 22529 Hamburg, Germany; hanna-lisa.goebels@uke.de (H.-L.G.); heinemann@uke.de (A.H.); b.ondruschka@uke.de (B.O.); 2Department of Intensive Care Medicine, University Medical Center Hamburg-Eppendorf, Martinistr. 52, 20246 Hamburg, Germany; k.roedl@uke.de (K.R.); d.jarczak@uke.de (D.J.); s.kluge@uke.de (S.K.); 3Department of Cardiology, Angiology and Intensive Care Medicine, Marien Hospital, 22087 Hamburg, Germany; bwzkkliniki@bundeswehr.org; 4Department of Pneumonology and Intensive Care Medicine, Weaning Center, Asklepios Hospital Barmbek, 22307 Hamburg, Germany; f.ludwig@asklepios.com; 5Department of Intensive Care and Respiratory Medicine, Clinic for Airway, Thorax and Respiratory Medicine, Asklepios Hospital Harburg, 21075 Hamburg, Germany; ma.bachmann@asklepios.com; 6Department of Anesthesiology, Intensive Care, Emergency Medicine and Pain Medicine, Asklepios Hospital St. Georg, 20099 Hamburg, Germany; b.bein@asklepios.com; 7Department of Anesthesiology, Intensive Care and Emergency Medicine, Asklepios Hospital Wandsbek, 22043 Hamburg, Germany; c.weber@asklepios.com; 8Department of Cardiology, Albertinen Hospital, 22457 Hamburg, Germany; karsten.sydow@immanuelalbertinen.de; 9Department of Internal Medicine, Bethesda Hospital Bergedorf, 21029 Hamburg, Germany; bota@bkb.info; 10Department for Anesthesiology and Intensive Care Medicine, Amalie Sieveking Hospital, 22359 Hamburg, Germany; hans.paschen@immanuelalbertinen.de; 11Department of Internal Medicine, Agaplesion Diakonie Hospital Hamburg, 20259 Hamburg, Germany; andreas.deweerth@d-k-h.de; 12Department of Interdisciplinary Intensive Care Medicine, Bundeswehr Hospital, 22049 Hamburg, Germany; carstenveit@hotmail.com; 13Department of Anesthesiology, Intensive Care, Emergency Medicine and Pain Medicine, Asklepios Hospital Nord, 22417 Hamburg, Germany; o.detsch@asklepios.com; 14Department of Anesthesiology and Intensive Care Medicine, Helios Mariahilf Hospital, 21075 Hamburg, Germany; anaesthesie.mariahilf@helios-gesundheit.de

**Keywords:** SARS-CoV-2, COVID-19, venous thromboembolism, deep vein thrombosis, pulmonary embolism, respiratory infections

## Abstract

Critically ill COVID-19 patients are at high risk for venous thromboembolism (VTE), namely deep vein thrombosis (DVT) and/or pulmonary embolism (PE), and death. The optimal anticoagulation strategy in critically ill patients with COVID-19 remains unknown. This study investigated the ante mortem incidence as well as postmortem prevalence of VTE, the factors predictive of VTE, and the impact of changed anticoagulation practice on patient survival. We conducted a consecutive retrospective analysis of postmortem COVID-19 (*n* = 64) and non-COVID-19 (*n* = 67) patients, as well as ante mortem COVID-19 (*n* = 170) patients admitted to the University Medical Center Hamburg-Eppendorf (Hamburg, Germany). Baseline patient characteristics, parameters related to the intensive care unit (ICU) stay, and the clinical and autoptic presence of VTE were evaluated and statistically compared between groups. The occurrence of VTE in critically ill COVID-19 patients is confirmed in both ante mortem (17%) and postmortem (38%) cohorts. Accordingly, comparing the postmortem prevalence of VTE between age- and sex-matched COVID-19 (43%) and non-COVID-19 (0%) cohorts, we found the statistically significant increased prevalence of VTE in critically ill COVID-19 cohorts (*p* = 0.001). A change in anticoagulation practice was associated with the statistically significant prolongation of survival time (HR: 2.55, [95% CI 1.41–4.61], *p* = 0.01) and a reduction in VTE occurrence (54% vs. 25%; *p* = 0.02). In summary, in the autopsy as well as clinical cohort of critically ill patients with COVID-19, we found that VTE was a frequent finding. A change in anticoagulation practice was associated with a statistically significantly prolonged survival time.

## 1. Introduction

The severe acute respiratory syndrome coronavirus 2 (SARS-CoV-2) emerged in 2019, causing a global healthcare emergency [1]. Up to 20% of patients with coronavirus disease 2019 (COVID-19) became hospitalized, and up to 5% were admitted to the intensive care unit (ICU) [2,3,4]. Although different treatment methods and therapeutic strategies have been proposed, mortality rates in critically ill patients remain unacceptably high [2,5].

Compared to other respiratory infections, COVID-19 is associated with an increased incidence of thrombotic events and disseminated intravascular coagulation (DIC) [6,7,8,9]. Venous thromboembolism (VTE), namely deep vein thrombosis (DVT) and/or pulmonary embolism (PE), was found in up to 31% of critically ill COVID-19 patients clinically [8]. Mechanistically, micro- and macrovascular thrombosis might result from endothelial dysfunction and coagulation activation, leading to thromboinflammatory states [9,10,11,12]. Different markers, including C-reactive protein and D-dimers, have been identified as predictive of disease severity as well as VTE [7,11].

Prevention and recognition of VTE are of significant interest, especially in patients suffering from severe illness. Despite standard anticoagulation regimes, the burden of thrombotic events remained high. Consequently, extended measures for the prophylaxis of venous thromboembolism have been introduced [13,14]. In the federal state of Hamburg (Hamburg, Germany), the recommendations for prophylactic anticoagulation were adapted on 5 May, 2020, following a small case series revealing VTE in 58% of critically ill COVID-19 patients. Of those, one-third died in a direct causal connection with pulmonary embolisms [6,14]. In detail, this change in anticoagulation practice comprised an increase in dosage in low-molecular-weight heparin from a prophylactic to an intermediate-dose regime in patients without signs of anticoagulation disorders.

Several studies have attempted to conduct benefit–risk assessments of different prophylactic anticoagulation regimes in critically ill COVID-19 patients [15,16]. Nevertheless, their effectiveness in reducing VTE incidence remains unclear. This multicentre investigation aimed to investigate the postmortem prevalence as well as incidence of VTE, the factors predictive of VTE, and the impact of changed anticoagulation practice on patient survival.

## 2. Materials and Methods

### 2.1. Study Design and Setting

Consecutive retrospective analysis of postmortem and ante mortem COVID-19 patients admitted to the University Medical Center Hamburg-Eppendorf (Hamburg, Germany) between 1 March and 31 December, 2020. The ethics committee of the Hamburg Chamber of Physicians approved the study (No.: 2020-10353-BO-ff and PV7311).

### 2.2. Study Cohorts

#### 2.2.1. Postmortem Cohorts

##### COVID-19 Autopsy Cohort 

All consecutive autopsied adult COVID-19 decedents, treated at intensive care units all over the state of Hamburg, admitted to the Institute of Legal Medicine at the University Medical Center Hamburg-Eppendorf (Hamburg, Germany) were included between 1 March and 31 December, 2020 (*n* = 64). COVID-19 was defined by the medical history and ≥1 positive antemortem and/or postmortem reverse transcription-quantitative polymerase chain reaction (RT-qPCR) for SARS-CoV-2. According to an established death case evaluation, the final cause of death was attributed to COVID-19 in all cases [17].

##### Non-COVID-19 Autopsy Cohort 

All consecutive autopsied adult decedents treated at the Department of Intensive Care Medicine (University Medical Center Hamburg-Eppendorf, Hamburg, Germany) and consecutively admitted to the Institute of Legal Medicine (University Medical Center Hamburg-Eppendorf, Hamburg, Germany) were included between 1 January, 2019, and 29 February, 2020 (*n* = 131). Deceased from intensive care units other than the Department of Intensive Care Medicine were excluded due to the availability of ante mortem data.

##### Postmortem External and Internal Examination

External and internal examinations were performed according to guidelines on postmortem examinations by the German Society of Legal Medicine (AWMF 054/001) with special consideration of the guidelines on handling COVID-19 deaths. Photographic and written documentation was performed in all cases. The postmortem interval was 4 days (IQR: 2–6) on the median.

##### Postmortem Diagnosis of Venous Thromboembolism

The postmortem diagnosis of venous thromboembolism was based on the patients’ medical records and external and internal corpse examinations. Blood clots in the pulmonary arteries and/ or lower leg veins were analyzed regarding their temporal nature. Blood clots were defined acutely based on their surface and structure in macroscopic cross-sections (i.e., clots of ribbed structure without relevant adhesions to the vessel walls).

##### Clinical Data Collection—Postmortem Cohort

Clinical data were obtained from discharge letters provided by the hospitals responsible for the patients’ medical treatment. Furthermore, a paper-based case report form was sent to the clinics requesting additional clinical data, including organ support, specific COVID-19 treatment, and prophylactic anticoagulation strategy within the last 48 h prior to death.

#### 2.2.2. Ante Mortem Cohort

##### Ante Mortem COVID-19 Intensive Care Unit Cohort

All consecutive critically ill adult COVID-19 patients admitted to the Department of Intensive Care Medicine (University Medical Center Hamburg-Eppendorf, Hamburg, Germany) were included between 1 March, 2020, and 31 December, 2020 (*n* = 170). COVID-19 was defined by the patients’ clinical features and ≥1 positive RT-qPCR for SARS-CoV-2.

##### Microbiological Examination

Respiratory tract samples (i.e., tracheal aspirates and nasopharyngeal swabs (eSwab, Copan, Italy)) and blood plasma (Serum-Gel, Sarstedt, Nümbrecht, Germany) were obtained as part of the clinical routine at the Department of Intensive Care Medicine. SARS-CoV-2 RNA was detected and quantified as described previously [18]. Standard RNA reference material (obtained from INSTAND e.V., Duesseldorf, Germany) was used for quantification.

##### Ante Mortem Diagnosis of Venous Thromboembolism

The ante mortem diagnosis of venous thromboembolism (i.e., pulmonary embolisms and/or deep vein thrombosis) was based on clinical suspicion and the subsequent confirmatory diagnostic procedures. Clinical risk assessment for VTE was conducted in accordance with recently published guidelines [19]. In clinical suspicion, confirmatory ultrasound and computed tomography pulmonary angiography were performed according to the local standard operating procedures.

##### Clinical Data Collection—Ante Mortem Cohort

Clinical data were collected through the electronic patient data management system (PDMS, Integrated Care Manager^®^ (ICM), Version 9.1—Draeger Medical, Lübeck, Germany). The covariables included were age, sex, comorbidities, COVID-19 disease specifics (date of diagnosis, severity, and complications), length of ICU stay, length of hospital stay, survival status, anticoagulation modalities (within the last 48 h prior to discharge), organ support (i.e., mechanical ventilation, vasopressor, and renal replacement therapy) and laboratory parameters. 

#### 2.2.3. Anticoagulation Regimes

Patients were grouped according to anticoagulation regimes received within 24 h before death, i.e., low, intermediate, and high anticoagulation dose. A low anticoagulation dose was defined as a prophylactic regimen of enoxaparin (or a similar substance) at a dose of 20 mg/40 mg once daily. The intermediate anticoagulation dose was defined as 30 mg/40 mg subcutaneously twice daily (weight-based enoxaparin 0.5 mg/kg). Alternatively, unfractionated heparins aiming at 1.5- to 1.8-fold prolonged partial thromboplastin time (PTT) were counted as intermediate-dose anticoagulation. A high anticoagulation dose was defined as a therapeutic dose of enoxaparin (or a similar substance) at a dose of 1 mg/kg subcutaneously twice daily if creatinine clearance (CrCl) was 30 mL/min/1.73 m^2^ or higher or 0.5 mg/kg twice daily if CrCl was 15–29 mL/min/1.73 m^2^. Furthermore, patients with continuous anticoagulation receiving unfractionated heparins or argatroban, aiming at a PTT of 60 s or higher, were included in this group.

#### 2.2.4. Statistical Analysis

Data are presented as absolute and relative frequencies for categorical variables and as median and interquartile range for continuous variables. As appropriate, the comparison of categorical variables was made using Fisher’s exact or chi-square test. Continuous variables were compared using the Mann–Whitney U-test. Cases and controls for comparative autopsy cohorts were matched randomly based on age and sex. Univariate and multivariate comparison was made by logistic regression. Independent variables were included on a clinical and scientific basis. Odds ratios with 95% confidence intervals are reported. Model fitness was estimated using likelihood ratio (LR) chi2 testing and Akaike’s information criterion (AIC). Survival function estimates were calculated using the Kaplan–Meier method and were compared using the log-rank test. Values of *p* < 0.05 were considered statistically significant. Statistical analyses were performed using Stata/MP 17.0 (StataCorp, College Station, TX, USA). GraphPad Prism software version 9.0.0 (GraphPad Software, San Diego, CA, USA) was used for data illustration.

## 3. Result

### 3.1. Study Population

In the study period from 1 March to 31 December, 2020, 632 COVID-19-related deaths occurred in the city of Hamburg (Hamburg, Germany) officially [17]; all cases were evaluated at the Institute of Legal Medicine of the University Medical Center Hamburg-Eppendorf. Of those, we could identify 64 consecutive critically ill patients who were treated at an ICU and were evaluated by full autopsy and medical records. The median age at death was 73 (IQR: 64—79) years; 27% (*n* = 17) were female. Patient characteristics are illustrated in Table 1.

### 3.2. Postmortem Prevalence of VTE in Consecutive COVID-19 Decedents Compared to Non-COVID-19 Decedents

In consecutive autopsied COVID-19 cases, 24/64 (38%) cases were found to have VTE. In detail, PE was found in 19/64 (30%) patients, and DVT was found in 18/37 (49%) patients. Of the pulmonary embolisms, five were localized in central, eight in segmental, and six in subsegmental flow areas of the pulmonary vasculature. In 6/19 (32%) cases, the pulmonary embolism remained clinically undiagnosed. 

To distinguish the occurrence of venous thromboembolism in COVID-19 patients from that in critically ill patients in general, their occurrence was studied in an age- and sex-matched subset cohort (21 COVID-19 patients and 21 non-COVID-19 patients). The latter were included from a consecutive autopsied cohort of non-COVID-19 patients (1 January 2019–29 February 2021) (cohort specifications are shown in detail in Appendix A). VTE was found in 0/21 (0%) and 9/21 (43%) critically ill non-COVID-19 and COVID-19 patients, respectively (X^2^ = 11.45, *p* = 0.001). In the COVID-19 cohort, PE could be detected in 9/21 (43%) patients and DVT in 7/14 (50%). In the non-COVID-19 cohort, neither patients with PE nor DVT could be detected.

Clinical characteristics of decedents with and without PE were compared by uni- and multivariate analyses (see Table 1). No differences were found in regard to patient age (*p* = 0.13), sex (*p* = 0.80), or body mass index (BMI) (*p* = 0.53). Similarly, no differences were found with regard to the patients’ pre-existing medical conditions, presence of ARDS, or establishment of COVID-19-related therapies. Renal replacement therapy (RRT) was significantly associated with the occurrence of PE in patients with COVID-19 (OR: 11.22 [95% CI 2.36–53.30]; *p* = 0.01). Mechanical ventilation associates with PE in critically ill COVID-19 patients in univariate (OR: 3.09 [95% CI 0.62–15.42]; *p* = 0.02) but not multivariate (*p* = 0.44) analyses. 

Relevant laboratory parameters were compared with regard to their predictive function for the occurrence of pulmonary embolisms. Laboratory analyses revealed no statistically significant differences between patients with and without PE in leukocyte counts (13.8 × 109/L vs. 12.9 × 109/L, *p* = 0.78) and platelet count (175.0 × 109/L vs. 155.0 × 109/L, *p* = 0.57) before death, nor in the maximum levels of D-dimers (6.7 mg/L vs. 8.0 mg/L, *p* = 0.32) or C-reactive protein (184.0 mg/dL vs. 172.0 mg/dL, *p* = 0.81).

### 3.3. Effect of Local Guideline Changes on Anticoagulant Therapy

Following initial reports of a high prevalence of thrombotic events, local guidelines on anticoagulant therapy for hospitalized patients in the city of Hamburg were adjusted by May 2020 [6]. A statistically significant prolongation of survival time of critically ill COVID-19 patients was observed for cases between May and December 2021 compared to the period before, as confirmed by the log-rank test (HR: 2.55 [95% CI 1.41–4.61], *p* = 0.01) (see Figure 1). A change in clinical practice, associated with less frequent occurrence of VTE (54% vs. 25%; X^2^ = 5.49, *p* = 0.02) and PE (39% vs. 22%; X^2^ = 2.20, *p* = 0.14).

### 3.4. Ante Mortem Clinical Incidence, the Likelihood of Death, and the Effect of Anticoagulant Regimens on the Occurrence of VTE in Consecutive Critically Ill COVID-19 Patients Admitted to an ICU

To estimate the incidence of VTE in critically ill COVID-19 patients, a consecutive cohort was clinically screened for the occurrence of venous thromboembolism (*n* = 170). The median age was 63 years (IQR: 55–73), and 34% (*n* = 58) were female. Patients survived COVID-19 in 99/170 (58%) cases and died in 71/170 (42%) cases.

Of those, 29/170 (17%) suffered VTE diagnosed during their lifetime. In detail, 12/170 (7%) suffered PE, and 18/170 (11%) suffered DVT. Location of PE was central, segmental, and subsegmental in two, eight, and two cases, respectively. On a side note, performing subgroup analyses of all patients autopsied, 12/23 (52%) patients suffered VTE as confirmed by full autopsy (PE: 10/23 [44%], DVT: 9/22 [41%]). Importantly, 9/12 (75%) died of pulmonary embolisms; the likelihood of death in the presence of PE was estimated for this cohort (HR: 1.77 [95% CI 0.88–3.57], *p* = 0.11).

Multivariate analyses confirmed the need for ECMO (OR: 41.33 [95% CI 5.54–308.31], *p* < 0.0001) and renal replacement therapy (OR: 10.55 [95% CI 2.59–43.02], *p* = 0.001) as strong predictors of patient death (see Table 2).

Next, survivors and non-survivors with (*n* = 12) and without (*n* = 158) pulmonary embolisms were compared by univariate analyses. Diabetes mellitus (67% vs. 31%, *p* = 0.01) was significantly associated with the presence of PE. Further the severity of ARDS (median: 3 (3–3) vs. median: 2 (0–3), *p* = 0.04), time of mechanical ventilation (median: 19 [5–58] days vs. median: 8 [0–20] days, *p* = 0.04) and need for RRT (100% vs. 42%, *p* < 0.0001) were associated with the occurrence of PE (see Table 3).

Relevant laboratory parameters were investigated with regard to their predictive function for the patients’ survival. The peak leukocyte counts (19.9 × 109/L vs. 14.1 × 109/L, *p* = 0.003), D-dimer levels (16.3 mg/L vs. 4.74 mg/L, *p* < 0.0001), C-reactive protein levels (279.0 mg/dL vs. 194.0 mg/dL, *p* < 0.0001), and procalcitonin levels (7.1 ng/mL vs. 0.8 ng/mL, *p* < 0.0001) were shown to be associated with fatal outcomes. Likewise, low platelet counts were associated with fatal outcomes (63.0 × 109/L vs. 164.0 × 109/L, *p* < 0.0001). Laboratory parameters upon ICU admission were not predictive of the patients’ survival. Presence of PE was associated with D-dimer levels (9.21 mg/L vs. 2.91 mg/L, *p* = 0.02) upon ICU admission. In addition, no association of laboratory parameters upon ICU admission nor of the peak parameters with the occurrence of pulmonary embolisms in critically ill COVID-19 patients was found (see Figure 2).

We found SARS-CoV-2 RNA in the blood of 109/155 (70%) patients. Aiming to investigate the relationship between the occurrence of PE and a measurable level of viremia in critically ill COVID-19 patients, a statistically significant inverse association of viremia with the occurrence of pulmonary embolism was found (73% vs. 42%; X^2^ = 5.12, *p* = 0.02).

Next the influence of different doses of anticoagulant therapy on the occurrence of PE was investigated. Thereby, we found that anticoagulation in a dose-dependent manner was not statistically significant associated with a reduced incidence of pulmonary embolism (X^2^ = 6.42, *p* = 0.13). Yet, patients receiving prophylactic (1/30 (3%)) and intermediate-dose (0/31 (0%)) anticoagulation presented with low frequencies of PE. Meanwhile, in patients receiving therapeutic dose anticoagulation, relatively high frequencies of PE were observed (11/12 [92%]). In total, 11 patients did not receive anticoagulant therapy due to bleeding complications (n = 4) or thrombocyte counts < 50/µL (*n* = 7). Notably, in a multivariate model, corrected for the occurrence of PE (OR: 2.21 [95% CI 0.40–12.08], *p* = 0.36), anticoagulant regimes with low-molecular-weight heparin (low dose, OR: 0.08 [95% CI 0.02–0.43], *p* = 0.003; intermediate dose, OR: 0.06 [95% CI 0.01–0.29], *p* = 0.001; high dose, OR: 0.06 [95% CI 0.01–0.28], *p* < 0.0001) but not unfractionated heparin (low dose, OR: 1.65 [95% CI 0.14–19.85], *p* = 0.69; high dose, OR: 0.39 [95% CI 0.10–1.61], *p* = 0.19) showed a statistically significant dose-dependent improvement of the patients’ survival (see Table 4).

## 4. Discussion

In this large observational postmortem and clinical analysis of critically ill patients with COVID-19, we found an elevated risk of thrombosis, which was impressively demonstrated by comparing the postmortem prevalence of VTE in COVID-19 and non-COVID-19 patients. Furthermore, the analysis found a statistically significant association of patient outcome with the change in anticoagulation practice. Interestingly, we were able to show that a majority of critically ill patients presented viremic during the ICU stay, which could contribute to an elevated thromboinflammatory state and immunothrombosis. 

Thromboembolic events commonly complicate the disease course of patients suffering from COVID-19 [6,8]. Accurate assessment of VTE in hospitalized patients with COVID-19 is difficult, and reported incidence rates range from 5 to 85% [10]. The variability of incidence could be a consequence of different factors, including testing strategy, clinical setting, and degree of thromboprophylaxis [20]. Patients hospitalized for any circumstances, but especially for COVID-19, generally face an elevated risk of thrombosis, especially critically ill patients, who are at the highest risk for thromboembolic events despite prophylactic anticoagulation [6,21]. This owes to different factors unique in the ICU, including prolonged immobilization, central venous catheterization, blood product transfusion, and certain medications [22]. One clinical observational study, including critically ill COVID-19 patients, revealed that up to 31% of the patients suffer from VTE, with 81% suffering pulmonary embolisms being the most commonly diagnosed thromboembolic complication [8]. The presence of VTE was associated with increased illness severity and mortality [23,24]. In our large cohort of 64 deceased COVID-19 patients from ICUs all over Hamburg, we could confirm earlier reports regarding the high incidence of VTE. Previous studies in the early phase of the pandemic revealed VTE in 7/12 (58%) and 32/80 (40%) of autopsied cases [6,25]. The postmortem evaluation of our cohort revealed that 30% had PE and 49% DVT. In comparison, in our clinical cohort, we found a much lower incidence of VTE. This may be explained by the fact that clinical diagnostics for VTE were only performed on reasonable clinical signs for VTE, as also underscored by previous prospective clinical cohort studies, revealing a high prevalence of occult thrombosis in mild to moderate COVID-19 cases [26]. Although the risk is known to be high, the exact incidence of VTE in critically ill patients is a matter of ongoing debates. Furthermore, data comparing the postmortem VTE incidence in COVID-19 and non-COVID-19 deceased critically ill patients remain scarce. To our knowledge, we are reporting the largest consecutive cohort of COVID-19 deceased from ICU solely who underwent autoptic evaluation. Further, we compared our data to a non-COVID-19 cohort of critically ill deceased. We retrospectively investigated the occurrence of PE and DVT, in an age and sex-matched cohort. In our analysis, only critically ill patients with COVID-19 suffered from VTE. Although our matched cohort was small, all patients were critically ill and underwent full-body examination using autopsy, representing the gold standard of postmortem diagnostics. The circumscribed low VTE prevalence in non-COVID-19 cases aligns with the VTE incidence previously described in ICU patients screened for VTE occurrence [27]. However, patients in the control cohort were heterogeneous and not only due to respiratory diseases in the intensive care unit. A prospective cohort investigating patients with COVID-19 and non-COVID-19-related ARDS showed a significantly higher rate of VTE in COVID-19 patients [28]. Our findings impressively underline the high frequency of VTE in critically ill patients with COVID-19 and the high demand for sufficient anticoagulation strategies in the ICU setting, especially in COVID-19 patients. 

Findings of small autopsy case series early in the pandemic pointed out a high risk of VTE in patients with COVID-19 [6,21]. These reports resulted in discussions of the optimal anticoagulation strategy in patients with COVID-19 and led to the implementation of different anticoagulation regimes within hospitalized patients. Furthermore, the clinically important question is whether enhanced or therapeutic-dose anticoagulation as thromboprophylaxis is able to prevent VTE without drastically increasing the risk of major bleeding. Given the high morbidity and mortality associated with severe COVID-19 and the concern that aspects of the disease are driven by micro-/macrothrombosis, many hospitals initiated aggressive anticoagulation protocols. Two studies investigated the use of intermediate-dose anticoagulation in patients with severe COVID-19 and did not find significant differences regarding the occurrence of thrombosis or death [15,29]. Randomized controlled trials investigating therapeutic anticoagulation in comparison to standard therapy in critically ill patients did not result in a greater probability of survival or organ-support-free days; a beneficial effect was only shown in one study investigating high-risk patients exhibiting elevated D-dimer levels [16,30,31,32]. However, to date, the effect of changing anticoagulation practice on the incidence of venous thromboembolism has not been studied in a real-world cohort. In the city of Hamburg (Hamburg, Germany), a change in treatment recommendations on anticoagulation practice was made in early May 2020 based on published autopsy data [6]. We here could show that change in anticoagulation practice is associated with a significant extension in patient survival time. However, newly inspected beneficial therapies (e.g., dexamethasone) and improvement in the general clinical care and practice of critically ill COVID-19 patients after a few months of the pandemic make it difficult to say whether this change is attributable only to a change in anticoagulation practice. Nevertheless, potential lung damage could have been avoided by the mitigation of widespread thrombosis and microangiopathy of pulmonary vessels as previously described [9]. 

Besides SARS-CoV-2 itself, a number of mechanisms are thought to contribute to the elevated VTE risk identified. Abnormal levels of proinflammatory cytokines have been found in COVID-19 patients resulting in increased systemic inflammation and a prothrombotic endothelial dysfunction state [33]. In critically ill patients, immobilization, use of central venous catheters, platelet activation, and mechanical ventilation can contribute to a prothrombotic state [34]. In particular, elevated D-dimer levels have been shown to be significantly associated with the occurrence of PE as well as increased disease severity [20,35,36,37]. We could confirm these findings in our clinical cohort; of note, we were able to show that even D-dimer levels on ICU admission were significantly associated with the occurrence of PE in critically ill COVID-19 patients. Furthermore, we observed that severity of ARDS, length of MV and need for RRT were significantly associated with the occurrence of PE. The association with severity of ARDS and length of mechanical ventilation is most likely of reversed causation with PE leading to a higher degree of respiratory failure. It remains unclear whether VTE is directly associated with the severity of disease or whether more severe disease causes thrombosis. 

Additionally, viremia was previously suggested to be associated with the severity of illness in critically ill COVID-19 patients [38,39,40,41,42,43]. Furthermore, the failure to clear the virus in COVID-19 patients with underlying hematologic malignancies was associated with a high risk of death [44,45]. In our clinical cohort, we could show that viremia was detectable in 70% of critically ill COVID-19 patients. One recent study found that blood SARS-CoV-2 viral RNA load was independently associated with unfavorable outcomes in a large cohort of critically ill COVID-19 patients [18]. Earlier studies mainly reported single-point measurements and their association with unfavorable outcomes but did not consider VTE occurrence. In our clinical cohort, we found that fewer patients with viremia developed pulmonary embolisms (73% vs. 42%, *p* = 0.02). These patients presented higher D-dimer, CPR, and PCT values than non-viremia patients reflecting a higher risk of VTE. However, it is of note that patients without viremia developed VTE significantly more often. One could argue that viremia is not necessarily associated with a thromboinflammatory state. Generally, this finding would be in contrast to the previous association of critical illness and occurrence of VTE. Due to the retrospective nature of the study and smaller sample size this finding has to be interpreted with caution and must be confirmed in larger studies. 

Our study is subject to some limitations. Due to the study’s retrospective nature, the diagnosis of venous thromboembolism was based on clinical suspicion in the ante mortem cohort. Prospective cross-sectional studies are needed to confirm our findings on the prevalence and determinants of VTE. Comparing the outcome of different modes of anticoagulation in antemortem cohorts, no data on adverse side effects were included, making further benefit–risk assessments difficult. Randomized clinical trials might provide further insights into the effects of different anticoagulation regimens regarding the prevalence of VTE and the survival of critically ill patients.

## 5. Conclusions

In this large autopsy cohort of critically ill patients with COVID-19, we found that VTE was a frequent finding, and the postmortem prevalence was significantly higher compared to non-COVID-19 ICU patients. A change in practice in anticoagulation management is statistically significantly associated with prolonged survival time.

## Figures and Tables

**Figure 1 viruses-14-00811-f001:**
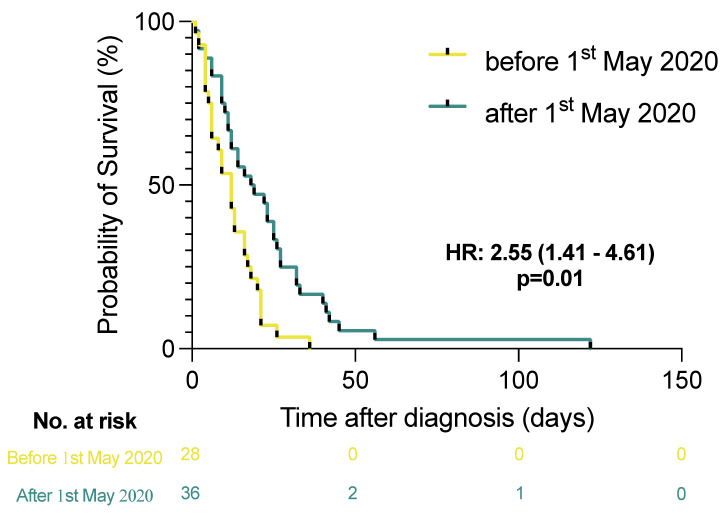
Kaplan–Meier curves display the overall time interval before death of 64 consecutive COVID-19 patients from intensive care units all over the city of Hamburg, autopsied at the Institute of Legal Medicine (University Medical Center Hamburg-Eppendorf, Germany). Survival time was calculated starting from the day of the first positive molecular genetic diagnosis (by RT-qPCR). Patients were grouped according to their admission date before or after adjustments to the local therapy recommendations (in early May 2020). Kaplan–Meier estimates and 95% confidence intervals, as well as hazard ratios (Mantel-Haenszel), are illustrated.

**Figure 2 viruses-14-00811-f002:**
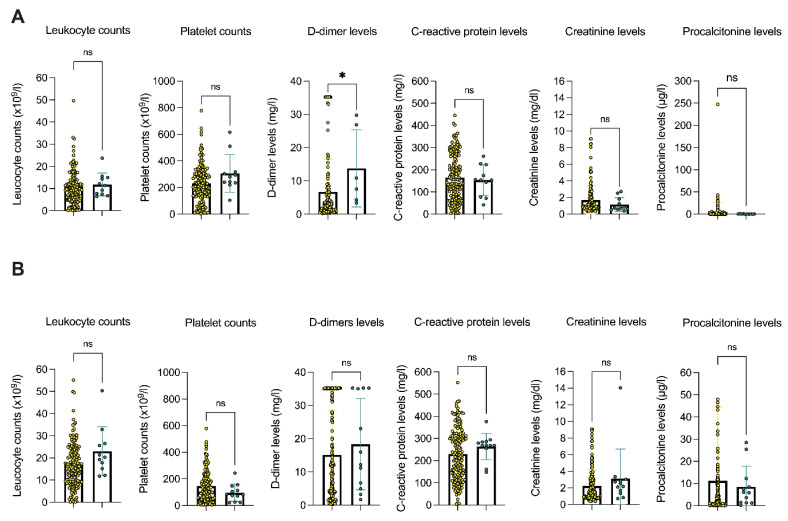
Relevant laboratory parameters of consecutive COVID-19 patients admitted to the Department of Intensive Care Medicine (University Medical Center Hamburg-Eppendorf, Germany) are illustrated (*n* = 170). Laboratory parameters at the time of admission (**A**) and peak parameters (**B**) are illustrated (only for platelet counts the minimum parameters are shown). Patients were grouped according to the presence of pulmonary embolisms (patients w/o pulmonary embolism are marked yellow on the left, and patients with pulmonary embolism are marked green on the right). Statistical comparisons between groups were performed using the Mann–Whitney U-test. *p*-values are displayed as follows: *, *p* < 0.05; ns, not significant.

**Table 1 viruses-14-00811-t001:** Baseline characteristics of consecutive COVID-19 patients from intensive care units all over the city of Hamburg, autopsied at the Institute of Legal Medicine (University Medical Center Hamburg-Eppendorf, Germany), are illustrated (*n* = 64). Patients were grouped according to the postmortem diagnosis of pulmonary embolisms. Numbers with frequencies and median with interquartile ranges are illustrated. Univariate and multivariate logistic regressions were performed. Odds ratios with 95% confidence intervals are given. Model estimators: Wald chi-square test: 17.29, *p* = 0.14; Akaike’s information criterion (AIC): 88.30. Abbreviations: BMI, body mass index; ICU, intensive care unit; ECMO, extracorporeal membrane oxygenation; RRT, renal replacement therapy; COVID-19, coronavirus disease 2019; ARDS, acute respiratory distress syndrome.

	Overall Patients	Pulmonary Embolism	Univariate Logistic Regression		Multivariate Logistic Regression	
			Odds ratio	*p*-value	Odds ratio	*p*-value
	*n* = 64	*n* = 19	95% CI		95% CI	
Sociodemographic variables						
Age	72.5 (63.5–79.0)	72.0 (57.0–79.0)	0.99 (0.94–1.03)	0.49	0.96 (0.91–1.01)	0.13
Sex (Ref: Female)	47 (73.4%)	14 (73.7%)	1.02 (0.30–3.44)	0.98	1.27 (0.21–7.49)	0.80
BMI	27.7 (22.6–32.8)	29.6 (24.7–36.8)	1.02 (0.98–1.07)	0.35	1.02 (0.96–1.09)	0.53
Pre-existing medical conditions						
Type II diabetes mellitus	18 (28.1%)	4 (21.1%)	0.59 (0.17–2.10)	0.42	0.49 (0.10–2.38)	0.38
Arterial hypertension	33 (51.6%)	9 (47.4%)	0.79 (0.27–2.31)	0.66	0.42 (0.08–2.12)	0.29
Chronic lung disease	20 (31.2%)	6 (31.6%)	1.02 (0.32–3.24)	0.97	0.96 (0.24–3.89)	0.95
Chronic kidney disease	13 (20.3%)	3 (15.8%)	0.66 (0.16–2.71)	0.56	0.56 (0.09–3.54)	0.54
ICU-related therapies						
Mechanical ventilation	50 (78.1%)	17 (89.5%)	3.09 (0.62–15.42)	0.02	2.89 (0.19–43.41)	0.44
ECMO	11 (17.2%)	3 (15.8%)	0.87 (0.20–3.70)	0.85	0.21 (0.04–1.13)	0.107
RRT	35 (54.7%)	15 (78.9%)	4.69 (1.34–16.46)	0.02	11.22 (2.36–53.30)	0.002
COVID-19-related therapies						
Remdesivir	1 (1.6%)	0 (0.0%)	ND	ND	ND	ND
Dexamethasone	19 (29.7%)	4 (21.1%)	0.53 (0.15–1.89)	0.33	0.76 (0.18–3.19)	0.71
Tocilizumab	0 (0%)	0 (0%)	ND	ND	ND	ND
COVID-19 disease severity						
ARDS	48 (75.0%)	15 (78.9%)	1.36 (0.38–4.93)	0.64	0.57 (0.09–43.63)	0.55

**Table 2 viruses-14-00811-t002:** Baseline characteristics of consecutive COVID-19 patients admitted to the Department of Intensive Care Medicine (University Medical Center Hamburg-Eppendorf, Germany) are illustrated (*n* = 170). Patients were grouped according to their overall survival. Numbers with frequencies and median with interquartile ranges are illustrated. Univariate and multivariate logistic regressions were performed. Odds ratios with 95% confidence intervals are given. Model estimators: Wald chi-square test: 64.96, *p* < 0.0001; Akaike’s information criterion (AIC): 140.73. Abbreviations: BMI, body mass index; ICU, intensive care unit; NIV, noninvasive ventilation; MV, mechanical ventilation; ECMO, extracorporeal membrane oxygenation; RRT, renal replacement therapy; COVID-19, coronavirus disease 2019; TPE, therapeutic plasma exchange; ARDS, acute respiratory distress syndrome; SAPS II, simplified acute physiology score 2; SOFA, sepsis-related organ failure assessment score.

	Overall Patients	Non-Survivors	Univariate Logistic Regression		Multivariate Logistic Regression	
			Odds ratio	*p*-value	Odds ratio	*p*-value
	*n* = 170	*n* = 71	95% CI		95% CI	
Sociodemographic variables						
Age	63.0 (55.0–73.0)	66.0 (58.0–76.0)	1.03 (1.00–1.05)	0.04	1.03 (0.98–1.08)	0.24
Sex (Ref: Female)	112 (65.9%)	49 (69.0%)	0.79 (0.41–1.50)	0.47	0.61 (0.16–2.33)	0.47
BMI	27.2 (24.2–31.9)	26.3 (23.9–32.7)	0.99 (0.95–1.04)	0.78	0.88 (0.80–0.96)	0.01
Charlson comorbidity index	2.0 (1.0–3.0)	2.0 (1.0–4.0)	1.09 (0.96–1.24)	0.17	1.50 (1.02–2.21)	0.04
Pre-existing medical conditions						
Type II diabetes mellitus	57 (33.5%)	22 (31.0%)	0.82 (0.43–1.57)	0.55	0.69 (0.19–2.57)	0.58
Arterial hypertension	97 (57.1%)	42 (59.2%)	1.16 (0.62–2.15)	0.64	1.75 (0.39–7.89)	0.47
Chronic lung disease	24 (14.1%)	11 (15.5%)	1.21 (0.51–2.89)	0.66	0.47 (0.12–1.95)	0.30
Chronic kidney disease	27 (15.9%)	10 (14.1%)	0.79 (0.34–1.85)	0.59	0.15 (0.01–2.05)	0.16
ICU-related therapies						
Nasal oxygen therapy	60 (35.3%)	24 (33.8%)	0.89 (0.47–1.69)	0.73	0.54 (0.12–2.44)	0.43
NIV	41 (24.1%)	19 (26.8%)	1.28 (0.63–2.59)	0.50	0.79 (0.13–4.62)	0.79
MV	120 (70.6%)	65 (91.5%)	8.67 (2.43–21.87)	<0.0001	0.98 (0.03–34.31)	0.99
MV time (days)	8.0 (0.0–21.5)	12.0 (5.0–22.0)	1.00 (0.98–1.01)	0.68	0.91 (0.87–0.96)	<0.0001
ECMO	49 (28.8%)	33 (46.5%)	4.50 (2.22–9.16)	<0.0001	41.33 (5.54–308.31)	<0.0001
RRT	79 (46.5%)	51 (71.8%)	6.47 (3.28–12.73)	<0.0001	10.55 (2.59–43.02)	0.001
Catecholamines	132 (77.6%)	67 (94.4%)	8.76 (2.94–26.08)	<0.0001	0.70 (0.03–16.54)	0.82
COVID-19-related therapies						
Remdesivir	32 (18.8%)	10 (14.1%)	0.57 (0.25–1.30)	0.18	0.49 (0.10–2.46)	0.38
Dexamethasone	73 (42.9%)	35 (49.3%)	1.56 (0.84–2.89)	0.16	1.48 (0.29–7.65)	0.64
Tocilizumab	3 (1.8%)	3 (4.2%)	ND	ND	ND	ND
TPE	6 (3.5%)	3 (4.2%)	1.41 (0.28–7.21)	0.68	2.37 (0.38–14.71)	0.36
COVID-19 disease severity						
ARDS	113 (66.5%)	64 (90.1%)	9.33 (3.89–22.36)	<0.0001	7.59 (0.27–211.19)	0.23
ARDS severity	3.0 (0.0–3.0)	3.0 (3.0–3.0)	2.29 (1.70–2.08)	<0.0001	1.79 (0.54–5.96)	0.34
SAPS II—on admission	40.0 (33.0–48.0)	43.0 (37.0–52.0)	1.07 (1.03–1.10)	<0.0001	1.05 (0.98–1.13)	0.17
SOFA score—on admission	7.0 (3.0–12.0)	10.0 (6.0–13.0)	1.12 (1.04–1.19)	0.001	0.92 (0.79–1.07)	0.27

**Table 3 viruses-14-00811-t003:** Baseline characteristics of consecutive COVID-19 patients admitted to the Department of Intensive Care Medicine (University Medical Center Hamburg-Eppendorf, Germany) are illustrated (*n* = 170). Patients were grouped according to the clinical diagnosis of pulmonary embolisms. Numbers with frequencies and median with interquartile ranges are illustrated. Abbreviations: BMI, body mass index; ICU, intensive care unit; NIV, noninvasive ventilation; MV, mechanical ventilation; ECMO, extracorporeal membrane oxygenation; ILA, interventional lung assist; RRT, renal replacement therapy; COVID-19, coronavirus disease 2019; TPE, therapeutic plasma exchange; ARDS, acute respiratory distress syndrome; SAPS II, simplified acute physiology score 2; SOFA, sepsis-related organ failure assessment score.

	No Pulmonary Embolism	Pulmonary Embolism	Overall Patients	Comparative Statistics (*p*-Value)
	*n* = 158	*n* = 12	*n* = 170	
Sociodemographic variables				
Age	63.0 (53.0–73.0)	62.5 (59.0–74.5)	63.0 (55.0–73.0)	0.54
Sex				
Male	104 (65.8%)	8 (66.7%)	112 (65.9%)	0.95
Female	54 (34.2%)	4 (33.3%)	58 (34.1%)	.
BMI	27.2 (24.2–31.9)	27.0 (23.7–31.8)	27.2 (24.2–31.9)	0.66
Charlson comorbidity index	2.0 (1.0–3.0)	2.5 (0.5–4.5)	2.0 (1.0–3.0)	0.58
Pre-existing medical conditions				
Type II diabetes mellitus	49 (31.0%)	8 (66.7%)	57 (33.5%)	0.01
Arterial hypertension	91 (57.6%)	6 (50.0%)	97 (57.1%)	0.61
Chronic lung disease	22 (13.9%)	2 (16.7%)	24 (14.1%)	0.79
Chronic kidney disease	24 (15.2%)	3 (25.0%)	27 (15.9%)	0.37
ICU-related therapy				
Nasal oxygen therapy	56 (35.4%)	4 (33.3%)	60 (35.3%)	0.88
NIV	36 (22.8%)	5 (41.7%)	41 (24.1%)	0.14
MV	109 (69.0%)	11 (91.7%)	120 (70.6%)	0.10
MV time	8.0 (0.0–19.5)	18.5 (4.5–58.0)	8.0 (0.0–21.5)	0.04
ECMO	45 (28.5%)	4 (33.3%)	49 (28.8%)	0.72
RRT	67 (42.4%)	12 (100.0%)	79 (46.5%)	<0.0001
Catecholamines	121 (76.6%)	11 (91.7%)	132 (77.6%)	0.23
COVID-19-related therapy				
Remdesivir	30 (19.0%)	2 (16.7%)	32 (18.8%)	0.84
Dexamethasone	67 (42.4%)	6 (50.0%)	73 (42.9%)	0.61
Tocilizumab	3 (1.9%)	0 (0.0%)	3 (1.8%)	0.63
TPE therapy	6 (3.8%)	0 (0.0%)	6 (3.5%)	0.49
COVID-19 disease severity				
ARDS	55 (34.8%)	2 (16.7%)	57 (33.5%)	0.20
ARDS severity	2.0 (0.0–3.0)	3.0 (3.0–3.0)	3.0 (0.0–3.0)	0.04
SAPSII	40.0 (33.0–48.0)	40.0 (34.0–52.5)	40.0 (33.0–48.0)	0.70
SOFA score	7.0 (3.0–12.0)	7.0 (2.5–11.5)	7.0 (3.0–12.0)	0.80

**Table 4 viruses-14-00811-t004:** Anticoagulation regimes of consecutive COVID-19 patients admitted to the Department of Intensive Care Medicine (University Medical Center Hamburg-Eppendorf, Germany) are illustrated (*n* = 170). Anticoagulation therapy was evaluated within the last 24 h prior to death. Patients were grouped according to their overall survival. Numbers with frequencies and median with interquartile ranges are illustrated. Univariate and multivariate logistic regressions were performed. Odds ratios with 95% confidence intervals are given. Model estimators: Wald chi-square test: 39.13, *p* < 0.0001; Akaike’s information criterion (AIC): 198.85.

	Overall Patients	Non-Survivors	Univariate Logistic Regression		Multivariate Logistic Regression	
	*n* = 170	*n* = 71	Odds ratio (95% CI)	*p*-value	Odds ratio (95% CI)	*p*-value
Pulmonary embolism	12 (7.1%)	9 (12.7%)	4.65 (1.21–17.83)	0.03	2.21 (0.40–12.08)	0.36
Low-molecular-weight heparin						
Low dose	23 (13.5%)	5 (7%)	0.14 (0.48–0.43)	<0.0001	0.08 (0.02–0.43)	0.003
Intermediate dose	31 (18.2%)	5 (7%)	0.10 (0.03–0.29)	<0.0001	0.06 (0.01–0.29)	0.001
High dose	31 (18.2%)	5 (7%)	0.10 (0.03–0.29)	<0.0001	0.06 (0.01–0.28)	<0.0001
Unfractionated heparin						
Low dose	7 (4.1%)	6 (8.5%)	14.06 (1.63–121.58)	0.02	1.65 (0.14–19.85)	0.69
High dose	56 (32.9%)	33 (46.5%)	3.36 (1.71–6.60)	<0.0001	0.39 (0.10–1.61)	0.19
Factor-IIa antagonist	9 (5.3%)	7 (9.9%)	5.30 (1.07–26.35)	0.04	0.84 (0.10–6.81)	0.87

## Data Availability

Data sharing is not applicable to this article.

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
