# Peer review of "New Insights in the Occurrence of Venous Thromboembolism in Critically Ill Patients with COVID-19—A Large Postmortem and Clinical Analysis"

_viruses, 2022, doi:10.3390/v14040811_

Round 1
Reviewer 1 Report
Thank you for this interesting and important study as post-mortem studies performed in patients that died from COVID19 are rare. There are numerous studies that reported on COVID19 infection and VTE in critically ill and non-critically ill COVID19 patients but most do not have post-mortem data, which is the true reflection of the incidence of VTE. In addition, the change in anti-coagulation practice mid-way in the study helped to illustrate if the addition of anti-coagulation changed the incidence of VTE and overall survival.
Clarifications on a few points would be helpful.
- There were 71 non-survivors reported in the ante mortem cohort, however, only 64 autopsies were performed. Is there a reason why 7 autopsies were missed out?
- How many of the patients that had post-mortem performed were on anti-coagulation before their demise?
- Were there any histological biopsies of the lung tissues and if so, were there any differences in the histological appearance of patients that were on anti-coagulation and those there were not. The prevalent theory is that the lung damage in COVID19 is due to thrombosis and microangiopathy of pulmonary vessels and it would be most interesting to see if the post mortem biopsies are able to show less thrombosis and microangiopathy in patients on anti-coagulation vs those without. If there is no difference, that would also be important to state.
Thank you.
Author Response
Reviewer 1
Thank you for this interesting and important study as post-mortem studies performed in patients that died from COVID19 are rare. There are numerous studies that reported on COVID19 infection and VTE in critically ill and non-critically ill COVID19 patients, but most do not have post-mortem data, which is the true reflection of the incidence of VTE. In addition, the change in anti-coagulation practice mid-way in the study helped to illustrate if the addition of anti-coagulation changed the incidence of VTE and overall survival.
Clarifications on a few points would be helpful.
- There were 71 non-survivors reported in the ante mortem cohort, however, only 64 autopsies were performed. Is there a reason why 7 autopsies were missed out?
R1: We thank the reviewer for this critical question. For conducting the present study, we used two consecutive cohorts of patients diagnosed with COVID-19. First, we report consecutive postmortem examined intensive care unit cases admitted to the Institute of Legal Medicine (University Medical Center Hamburg-Eppendorf), involving patients from more than 16 hospitals. Second, we report consecutive ante mortem intensive care unit cases admitted to the Department of Intensive Care Medicine (University Medical Center Hamburg-Eppendorf). We hope this clarifies the raised questions regarding the used cohorts. Please also compare lines 97 to 118 and 139 to 144.
- How many of the patients that had post-mortem performed were on anti-coagulation before their demise?
R2: We thank the reviewer for pointing this out. The majority of patients received any form of anticoagulation before their demise. However, some patients (n = 11) did not receive anticoagulant therapy in the last 24h before death. This was mainly due to bleeding complications (n = 4), or thrombocyte counts < 50/µl (n = 7). Please also compare lines 107 to 108.
- Were there any histological biopsies of the lung tissues and if so, were there any differences in the histological appearance of patients that were on anti-coagulation and those there were not. The prevalent theory is that the lung damage in COVID19 is due to thrombosis and microangiopathy of pulmonary vessels and it would be most interesting to see if the post mortem biopsies are able to show less thrombosis and microangiopathy in patients on anti-coagulation vs those without. If there is no difference, that would also be important to state.
R3: We thank the reviewer for this essential question. We agree that lung damage in patients with COVID-19 was attributable to thrombosis and microangiopathy in the lung, which was impressively demonstrated by Ackermann and colleagues (N Engl J Med 2020; 383:120-128). However, although of high interest, we could not provide this data because there was no systematic sampling of probes in the used cohort.
Thank you.

Reviewer 2 Report
Review: New insights in the occurrence of venous thromboembolism in critically ill patients with Covid-19- a large postmortem and clinical analysis.
Heinrich et al. have conducted a study where they have investigated the antemortem incidence and postmortem prevalence of VTE in ICU patients treated for Covid-19. They found an antemortem incidence similar to previous studies, but a postmortem prevalence significantly higher than the antemortem incidence. Furthermore, they conclude that a change of anticoagulation practice was associated with a prolonged the survival and reduced incidence of VTE in Covid-19 patients in the ICU. The high incidence of VTE in critically ill Covid-19 patients has been described and discussed thoroughly in previous papers, as the authos of the present paper also acknowledge. The findings of the significantly higher prevalence of VTE in the postmortem studies, however, are interesting and supports the use of intermediate doses of thromboprophylaxis in this patient population
I have some comments:
Major comments:
- That the incidence of VTE is increased in Covid-19 patients treated in the ICU is well known. Moreover, the postmortem prevalence is higher than the antemortem incidence, which has caused increased alertness about this condition.
With the autopsy, I am missing some discussion on the mechanism behind this phenomenon.
- Ante mortem- examined for VTE only upon clinical suspicion, no screening has been performed. A discussion addressing how this method has influenced the results is warranted.
- The prevalence of VTE in non-Covid postmortem population is low. However, this result should be compared to previous studies and discussed
Minor comments:
- The title refers to the study as large, however, number of patients (n) is not mentioned in the abstract. This should be provided.
- Abstract line 43—46: The sentence is not complete, something is missing.
- The numbering in the methods section should be reconsidered from 2.2.3
- The exact dosing regimens of LMWH should be described, not only low- intermediate and high.
Author Response
Reviewer 2
Review: New insights in the occurrence of venous thromboembolism in critically ill patients with Covid-19- a large postmortem and clinical analysis.
Heinrich et al. have conducted a study where they have investigated the antemortem incidence and postmortem prevalence of VTE in ICU patients treated for Covid-19. They found an antemortem incidence similar to previous studies, but a postmortem prevalence significantly higher than the antemortem incidence. Furthermore, they conclude that a change of anticoagulation practice was associated with a prolonged the survival and reduced incidence of VTE in Covid-19 patients in the ICU. The high incidence of VTE in critically ill Covid-19 patients has been described and discussed thoroughly in previous papers, as the authos of the present paper also acknowledge. The findings of the significantly higher prevalence of VTE in the postmortem studies, however, are interesting and supports the use of intermediate doses of thromboprophylaxis in this patient population
I have some comments:
Major comments:
- That the incidence of VTE is increased in Covid-19 patients treated in the ICU is well known. Moreover, the postmortem prevalence is higher than the antemortem incidence, which has caused increased alertness about this condition.
R4: We fully agree with the reviewer about the fact outlined, underscoring the importance of our findings.
With the autopsy, I am missing some discussion on the mechanism behind this phenomenon.
- Ante mortem- examined for VTE only upon clinical suspicion, no screening has been performed. A discussion addressing how this method has influenced the results is warranted.
R5: We thank the reviewer for this essential question. We agree that performing VTE diagnostics on clinical suspicion only might underestimate the true incidence, as previously demonstrated in prospective clinical cohort studies, revealing a high incidence of occult thrombosis in mild to moderate COVID-19 cases (Chen et al. International Journal of Infectious Diseases 2021).
- The prevalence of VTE in non-Covid postmortem population is low. However, this result should be compared to previous studies and discussed
R6: We thank the reviewer for raising this point and for the opportunity to clarify. Considering the size of the cohort included, the circumscribed prevalence of VTE in our non-COVID-19 cohort aligns with previous reports of VTE incidence ranging between 0.6 and 5.7%. We have specified on that in lines 491 to 493.
Minor comments:
- The title refers to the study as large, however, number of patients (n) is not mentioned in the abstract. This should be provided.
R7: We thank the reviewer for pointing this out. We have added the patient numbers accordingly (see lines 39 to 60). - Abstract line 43—46: The sentence is not complete, something is missing.
R8: We thank the reviewer for the comment, we have rephrased the sentence accordingly (see lines 43 to 46). - The numbering in the methods section should be reconsidered from 2.2.3
R9: We thank the reviewer for pointing this out and re-aligned this section starting from 2.2.3 (see lines 119 to 201). - The exact dosing regimens of LMWH should be described, not only low- intermediate and high.
R10: We thank the reviewer for pointing this out and added these details in the method section (see lines 167 to 187).
